# Functionalized MXene ink enables environmentally stable printed electronics

Tae Yun Ko[1,2,3,14], Heqing Ye[4,5,14], G. Murali[6,7,14], Seul-Yi Lee[8,14], Young Ho Park[6,7], Jihoon Lee [6,7], Juyun Lee[1,2,9], Dong-Jin Yun[10], Yury Gogotsi[11], Seon Joon Kim[1,2,12] ✉, Se Hyun Kim[5] ✉, Yong Jin Jeong[7,13] ✉, Soo-Jin Park[8] ✉ & Insik In[6,7] ✉

Establishing dependable, cost-effective electrical connections is vital for enhancing device performance and shrinking electronic circuits. MXenes, combining excellent electrical conductivity, high breakdown voltage, solution processability, and two-dimensional morphology, are promising candidates for contacts in microelectronics. However, their hydrophilic surfaces, which enable spontaneous environmental degradation and poor dispersion stability in organic solvents, have restricted certain electronic applications. Herein, electrohydrodynamic printing technique is used to fabricate fully solution-processed thin-film transistors with alkylated 3,4-dihydroxy-L-phenylalanine functionalized $Ti_3C_2T_x$ (AD-MXene) as source, drain, and gate electrodes. The AD-MXene has excellent dispersion stability in ethanol, which is required for electrohydrodynamic printing, and maintains high electrical conductivity. It outperformed conventional vacuum-deposited Au and Al electrodes, providing thin-film transistors with good environmental stability due to its hydrophobicity. Further, thin-film transistors are integrated into logic gates and one-transistor-one-memory cells. This work, unveiling the ligand-functionalized MXenes' potential in printed electrical contacts, promotes environmentally robust MXene-based electronics (MXetronics).

Solution processible two-dimensional (2D) materials as electrical contacts offer several crucial advantages over conventional vacuum-deposited metals, such as processing under ambient conditions, large-area fabrication, scalability, cost-effectiveness, compatibility with flexible substrates, and non-damaging to the underneath surface[1,2]. Titanium carbide MXene, denoted as $Ti_3C_2T_x$ (with $T_x$ representing −O, −OH, halogen, or chalcogen), has shown immense potential in various applications due to its 2D morphology and exceptional physico-chemical properties[3–10]. Its facile solution processability facilitates the seamless integration of controlled geometry thin films, simplifying architectural design and device manufacturing[1,11–16]. Chemically tunable electrical conductivity and high current-carrying capacity make MXenes well-suited for deployment as electrodes in microelectronics,

thin-film transistors (TFTs), and memory devices, defining a novel domain known as MXetronics[5,17–25]. However, the hydrophilic nature of MXenes remains a notable concern when considering their utilization in electronic circuits. These concerns primarily arise from two factors. Firstly, their propensity to interact with atmospheric water and oxygen raises apprehensions regarding the potential gradual deterioration of structural and compositional integrity, consequently impacting the electrical properties of MXene nanosheets[26]. Secondly, their hydrophilic tendencies contribute to poor dispersion stability in volatile organic solvents. This is crucial for avoiding undesirable coffee-ring effects during film deposition and necessitating low drying temperatures[27,28]. Notably, certain device fabrication processes are mandated to use such organic solvents[29].

Considering that the solution-processed films consist of an assembly of overlapping MXene nanosheets, the electron transport properties at the interface significantly influence the performance of MXene-based electronic devices. The presence of voids and inter-calants impair the inter-flake electron transport properties of MXene film. For instance, thin films of MXene nanosheets synthesized by the minimally intensive layer delamination (MILD) method, renowned for producing high-quality nanosheets, contain $Li^+$ ions as intercalants. These hydrophilic $Li^+$ ions facilitate the water molecules' penetration through the voids in the MXene film, entrapping them within the interlayer space[30,31]. Besides causing the degradation of MXene, the intercalated water may obstruct the efficient flow of charge carriers between the nanosheets in the MXene film. Consequently, MXene-based devices have yet to demonstrate stable performance under harsh atmospheric conditions[30]. Hence, the imperative challenge lies in modifying MXene nanosheet surface characteristics to achieve dispersion stability in volatile organic solvents, enhance oxidation resistance, mitigate moisture-induced swelling, and facilitate inter-flake electron transport. Tackling these challenges is crucial for MXenes' effective use in printed electronic devices.

Several strategies have been devised to modify MXene nanosheets, exploiting their surface terminal groups for interactions with metal ions, organic molecules, and polymers through hydrogen bonding, covalent bonding, and electrostatic interactions[31–37]. While some of these strategies enhance MXene dispersion in organic solvents, they often come at the cost of decreased electrical conductivity due to polymers or organic molecules obstructing inter-flake charge transport[32,36,38]. Recent efforts utilized catechol-based ligands to impart hydrophobic characteristics to MXene surfaces without sacrificing the essential electrical properties of MXene films[39]. Notably, alkylated 3,4-dihydroxy-L-phenylalanine (ADOPA) ligands, featuring a catechol-based head group and a hydrophobic tail that forms hydrogen bonds with MXene surfaces, enabled MXene nanosheet dispersion in volatile organic solvents such as ethanol and acetone[39]. The ADOPA-functionalized MXene (AD-MXene) films retained over 90% electrical conductivity (6,404 S cm$^{-1}$) of the pristine MXene films (6900 S cm$^{-1}$). Furthermore, benefitting from the hydrophobic nature of ADOPA ligands, AD-MXene films exhibited excellent oxidation stability and electrical conductivity retention under 85 °C at 85% relative humidity (RH) conditions for 20 days[39].

Here, we employ AD-MXene to print TFT circuit electrodes—source, drain, and gate. Electrohydrodynamic (EHD) printing technique was chosen for the TFTs fabrication through localized delivery of AD-MXene electrodes, zinc-tin-oxide (ZTO) active layer, and fluoro-co-hexafluoropropylene (PVDF-HFP)-based dielectrics. EHD printing offers advantages, including finer pattern features and compatibility with various materials and ink viscosities compared to conventional printing techniques[40–45]. However, EHD printing requires low-surface-tension solvents and cannot use MXenes dispersed in high-surface-tension water (72.8 dyne cm$^{-1}$)[29]. High-surface-tension solvents require a higher operating voltage, which is associated with voltage risks[29,46,47]. Ethanol's lower surface tension (22.0 dyne cm$^{-1}$) allows EHD printing at reduced voltages, making the excellent dispersion stability of AD-MXene in ethanol (up to 5 mg mL$^{-1}$) advantageous for producing MXene-based highly conductive electrodes[29]. Unlike the existing literature, the AD-MXene electrodes don't require complex post-fabrication strategies to avoid environmental degradation[48]. There is no need to add a binder to the ink, which is detrimental to the electrode conductivity, to enhance the electrode's adhesion and avoid the coffee-ring effects[28]. The performance of AD-MXene electrodes was compared with those of vacuum-deposited Au and Al electrodes. Furthermore, ADOPA ligands block moisture ingress, providing excellent environmental stability in TFTs with AD-MXene electrodes. They showed negligible changes in their transfer and output characteristics and bias

stabilities, even after 30 days of exposure to 60% RH at 25 °C. Finally, we assess AD-MXene's potential for integrated circuits, evaluating logic gates (complementary inverter, NAND, and NOR) and memory cells constructed using all-printed AD-MXene electrodes, ZTO active layers, and PVDF-HFP dielectrics.

## Results and discussion
### Properties of AD-MXene
As illustrated in Fig. 1a, the AD-MXene ink for EHD printing was prepared through facile mixing of aqueous dispersion of MXene with ethanol dispersion of ADOPA ligand at room temperature for around one hour and subsequent replacement of solvent with ethanol via centrifugation. The ADOPA ligand spontaneously adsorbed onto MXene due to its catechol head strong hydrogen-bonding and π-electron interactions with MXene surface functional groups, and the hydrophobic tail of ADOPA ligand enables the excellent dispersion of obtained AD-MXene in ethanol. The X-ray photoelectron spectroscopy (XPS) spectra for the Ti 2 $p$, O 1 $s$, C 1 $s$, and N 1 $s$ core levels of AD-MXene and pristine MXene are shown in Fig. 1b and Supplementary Fig. 1. Due to primary amine moiety in ADOPA, AD-MXene shows a clear peak around 399.9 eV in the N 1 $s$ region, confirming the attachment of ADOPA (Fig. 1b). To investigate further the interlayer structure and alignment of MXene sheets in the film, the X-ray diffraction (XRD) patterns of both pristine MXene and AD-MXene films were recorded as shown in Supplementary Fig. 2. The (002) peak shift from 7.32° to 6.38° was observed, corresponding to a $d$-spacing expansion from 1.22 nm to 1.36 nm, resulting from attached ligands. Compared to pristine MXene with poor dispersion properties in ethanol, AD-MXene shows excellent dispersion, as shown in Fig. 1c. The field emission scanning electron microscope (FE-SEM) image in Fig. 1d shows a representative AD-MXene flake from a batch with an average lateral size of 1.81 μm. Due to the enhanced hydrophobicity after ADOPA functionalization, our previous study has shown that AD-MXene is environmentally stable for an extended time, compared to pristine waterborne MXene, which is otherwise susceptible to water uptake and subsequent environmental degradation[39]. This has also been proven to be true in electronic devices with AD-MXene exposed to the surface, where a recent study shows that chemical sensors based on AD-MXene retained their performance for over 6 weeks of exposure in ambient environments[49].

### Electrical properties of EHD-printed AD-MXene electrodes
As shown in Fig. 2a, the as-prepared ethanol-based AD-MXene ink was loaded into EHD printing equipment and then ejected through the nozzle tip by applying an electric field between the nozzle and the substrate. The printing conditions were optimized by examining the jetting behavior and tuning the working distance (distance from the nozzle tip to the substrate) and the applied electric field. Four jetting behaviors, such as dripping, micro-dripping, cone-jet, and multi-jet, were noticed when the working distance and applied voltage were tuned in the ranges of 100–600 μm and 0.5–3.0 kV, respectively (Fig. 2b). The AD-MXene ink was printed into line-shaped patterns on a SiO$_2$/Si wafer at conditions optimized for uniform jetting in the stable cone-jet mode. These precise conditions encompassed a nozzle size of 185 μm, a printing speed of 10 mm s$^{-1}$, a controlled flow rate of 1.3 μL min$^{-1}$, and an applied voltage of 1.3 kV, all sustained at a working distance of 450 μm.

Optical microscope and cross-sectional FE-SEM images revealed the increased thickness of AD-MXene lines with increased printing cycles from 1 to 10 (Supplementary Figs. 3, 4). Further, atomic force microscopy (AFM) analysis revealed the gradual increase of average surface roughness ($R_a$) of AD-MXene lines with an increase in the number of printing cycles, and the $R_a$ was observed to be 7.8 nm for lines fabricated with 10 printing cycles (Supplementary Fig. 5). Characteristic current-voltage curves of AD-MXene electrodes

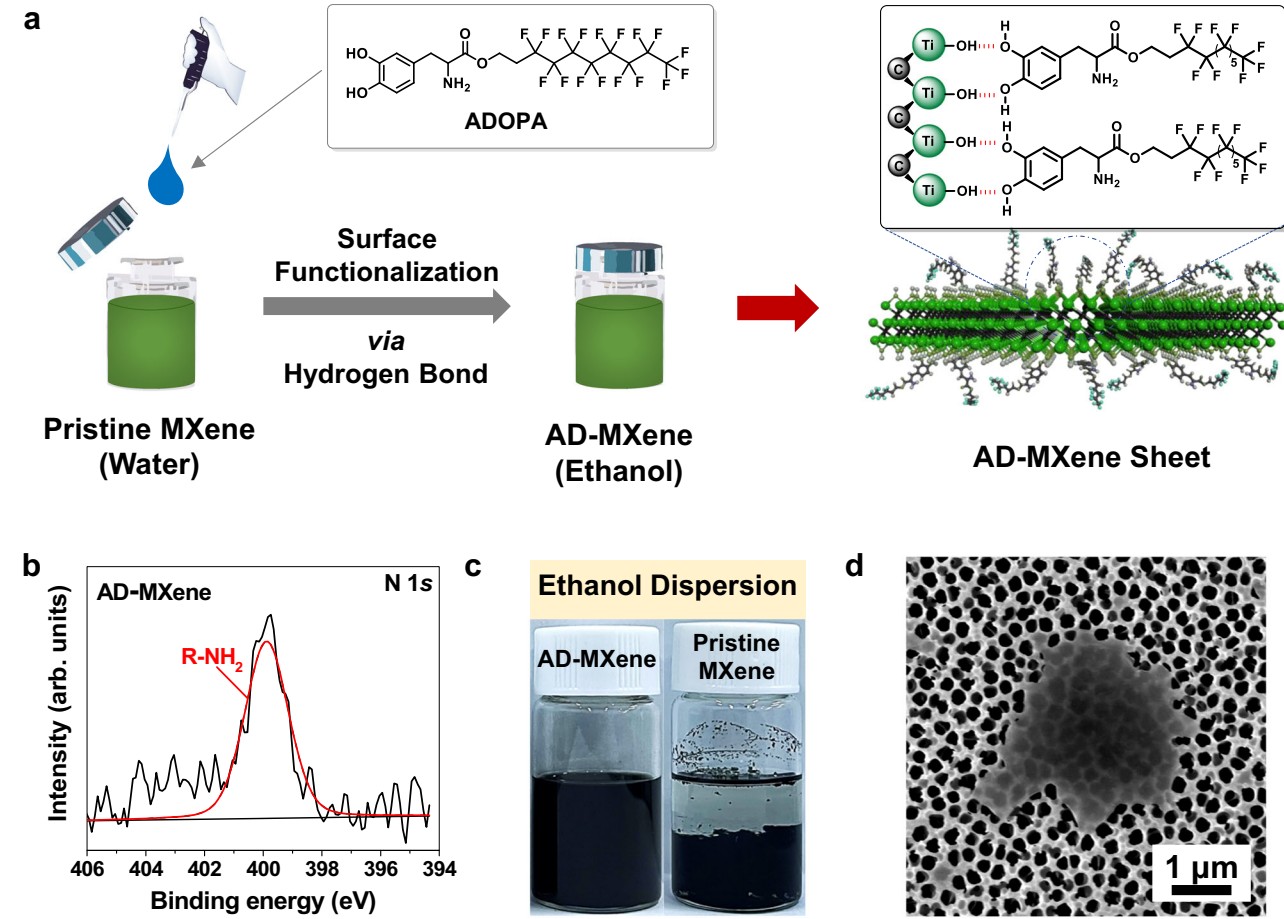

**Fig. 1 | Synthesis and characterization of AD-MXene. a** Schematic illustration of the general reaction procedure for preparing AD-MXene. **b** N 1$s$ XPS spectrum of AD-MXene. **c** Digital photograph of AD-MXene and pristine MXene dispersions in ethanol. **d** FE-SEM image of a single AD-MXene flake.

revealed that the increase in the thickness of lines (or number of printing cycles) enabled the enhanced current flow. In contrast, the increase in channel length decreased the current (Fig. 2c, d). Therefore, the EHD printing of AD-MXene with a controlled number of printing cycles and channel lengths may enable the facile fabrication of conductive wires/resistors of circuitries. The electrical conductivity of the AD-MXene electrode (length = 1 mm, width = 320 μm, and thickness = 190 nm) fabricated by 10 cycles of printing was observed to be 5579 S cm$^{-1}$, which was almost equal to that of the AD-MXene film fabricated using vacuum-assisted filtration. This is because the ADOPA ligands facilitated the alignment of AD-MXene flakes during the EHD printing process and enabled the printed AD-MXene electrodes to have excellent electrical conductivity (Supplementary Fig. 4). We hypothesize that the ligand's planar alignment and strong π-electron interaction with the MXene surface facilitated the inter-flake electron transport; as a result, the AD-MXene films exhibited electrical conductivity almost equal to that of pristine MXene film despite having ADOPA ligands as intercalants[39]. Notably, the conductivity of AD-MXene electrode (5579 S cm$^{-1}$) is considerably higher than previously reported poly(3,4-ethylenedioxythiophene)-poly(styrenesulfonate) electrodes (0.2–1200 S cm$^{-1}$)[50], reduced graphene oxide-based electrodes (590 S cm$^{-1}$)[51], and pristine MXene electrode (2600 S cm$^{-1}$)[23]. Besides, printing AD-MXene electrodes is more favorable as it doesn't require post-treatments, unlike other metal-based (Cu, Ag, Zn, etc.) inks[52–54]. The AD-MXene electrode designed in the square spiral pattern through continuous writing with the AD-MXene ink revealed the compatibility of AD-MXene ink for EHD printing of complex circuits (Fig. 2e).

## Performance of AD-MXene as electrodes of TFTs

The excellent compatibility of the AD-MXene ink for EHD printing and the high electrical conductivity of printed AD-MXene lines motivated us to investigate the effectiveness of AD-MXene as electrical contacts of TFTs. Therefore, we fabricated a TFT, wherein the AD-MXene was printed as source and drain electrodes on the ZTO active material deposited on SiO$_2$/Si substrate (Fig. 3a). ZTO was selected as an active layer due to its cost-effectiveness, solution processability, and potential as alternative to expensive indium-based counterparts[55]. Fig. 3b shows the transfer characteristics of TFTs prepared with AD-MXene electrodes deposited at different numbers of EHD printing cycles. All devices showed typical n-type transfer characteristics during ± 40 V of gate voltage ($V_G$) sweep (source to drain voltage ($V_D$) = 40 V). The increased number of AD-MXene electrode printing cycles enhanced the transfer performance of TFTs due to the improvement in electrical conductivity through the formation of denser network of AD-MXene electrode (Fig. 3b). The field-effect mobility (μ$_{FET}$) values were determined from $I_D^{1/2}$ vs. $V_G$ plots by following the equation: $I_D = μ_{FET}C_iW(2L)^{-1}(V_G - V_{th})^2$, where $I_D$, $C_i$, $W$, $L$, and $V_{th}$ are drain current, areal capacitance, channel width, channel length, and threshold voltage, respectively. The μ$_{FET}$ of devices gradually increased with an increase in the number of AD-MXene printing cycles or the thickness of MXene electrodes (Fig. 3c). For comparison, TFTs with vacuum-deposited Au and Al electrodes were also fabricated, and their performances were compared with that of AD-MXene devices (Fig. 3d–f). Figure 3e compares the transfer characteristics of ZTO-based TFT devices prepared with different source/drain electrodes (AD-MXene, Au, and Al). The ZTO TFT fabricated with AD-MXene source/drain

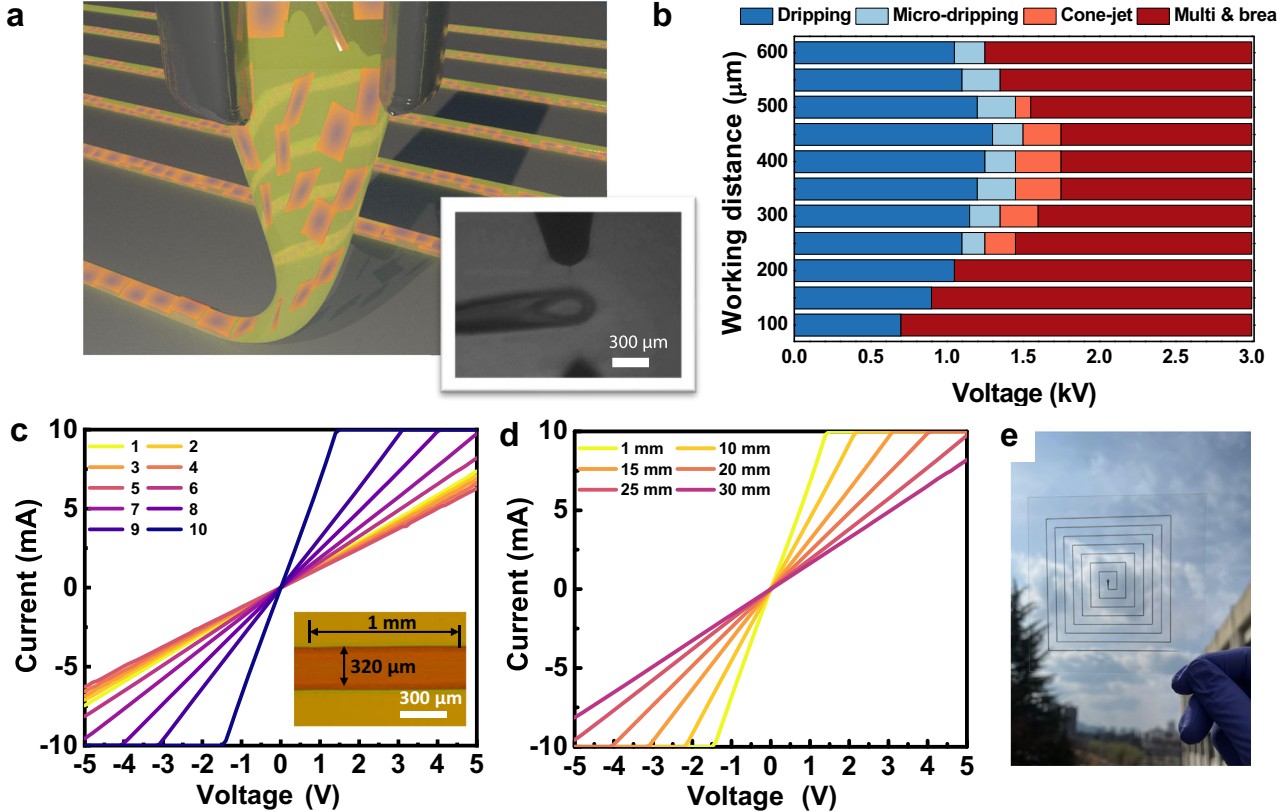

**Fig. 2 | EHD printing of AD-MXene electrodes and their current-voltage characteristics. a** Schematic showing the EHD printing of AD-MXene inks from a nozzle. The inset shows the optical microscope image of AD-MXene ink drops at the edge of the nozzle tip during the cone-jet mode of the EHD printing process. **b** Chart showing the EHD printing modes as a function of working distance and applied voltage. **c** Current–voltage characteristic curves of AD-MXene electrodes fabricated by 1–10 cycles of EHD printing. **d** Current-voltage characteristic curves of AD-MXene electrodes with 1–30 mm channel length (with 10 cycles of EHD printing). **e** View through an AD-MXene electrode printed in a square spiral geometry on glass.

electrodes exhibited better transfer characteristics than those fabricated with vacuum-deposited Al and Au electrodes. Hysteresis (the variation in the OFF-to-ON sweeping transfer curve compared to the ON-to-OFF transfer curve) was not observed for devices comprising Al and AD-MXene as source/drain electrodes. In contrast, the device with Au as source/drain electrodes exhibited poor electrical performance with hysteresis (Fig. 3e). Remarkably, the AD-MXene based TFT yielded considerably higher $\mu_{FET}$ of 3.24 cm$^2$ V$^{-1}$ s$^{-1}$ than those based on Al (2.61 cm$^2$ V$^{-1}$ s$^{-1}$) and Au (1.75 cm$^2$ V$^{-1}$ s$^{-1}$) electrodes (Fig. 3f). The off-current value of the AD-MXene based TFT was somewhat higher than that of the Au device. This was probably attributed to a combination of factors such as gate leakage, sub-threshold conduction, and trap states rather than the properties of the electrode material, all of which can be sufficiently optimized by device structure design and scale control of the printed electrode. Contact resistances of TFTs with different electrodes were extracted by the transfer length method from the width-normalized total resistance ($R_{tot}W$) of TFTs as a function of channel length shown in Supplementary Fig. 6. Although the $R_{tot}W$ showed a linear relationship with the channel length for all TFTs, AD-MXene electrodes caused low resistance as compared to Au and Al electrodes. Therefore, given that the crystalline morphology underneath the ZTO semiconductor layer was the same for all electrodes, the observed differences in $\mu_{FET}$ values of devices must have originated due to the changes in contact resistances at the interface of channel and electrode. In other words, we infer that the solution-processed AD-MXene electrodes provided an excellent interface with the ZTO active layer deposited through the same solution route compared to vacuum-deposited Au and Al electrodes.

## Performance of TFTs with different dielectric layers

The superiority and universality of AD-MXene electrodes was further demonstrated by utilizing them as gate, source, and drain electrodes in top-gate top-contact TFTs with PVDF-HFP and FPVDF-HFP dielectrics (Fig. 4). Insulating characteristics of PVDF-HFP and FPVDF-HFP dielectrics were evaluated before their integration in TFTs. To this end, metal-insulator-metal (MIM) capacitors with device structures of AD-MXene/PVDF-HFP/AD-MXene and AD-MXene/FPVDF-HFP/AD-MXene were fabricated by EHD-printing and their leakage current densities were measured (Supplementary Fig. 7a, b). The observed leakage current densities below 10$^{-8}$ A cm$^{-2}$ at an applied electric field strength of 4 MV cm$^{-1}$ for both devices revealed the excellent insulation properties of PVDF-HFP and FPVDF-HFP dielectrics. Furthermore, the areal capacitances of PVDF-HFP and FPVDF-HFP dielectrics at 1 kHz were measured to be 34.2 and 41.8 nF cm$^{-2}$, respectively (Supplementary Fig. 7c). The AD-MXene was coated on ZTO active layer as source/drain electrodes, after which PVDF-HFP or FPVDF-HFP was coated to form the dielectric layer. The EHD printing of AD-MXene as a gate electrode completed the fabrication of TFT with top-gate top-contact geometry, as the illustration and optical microscope images of devices shown in Fig. 4a. The transfer properties of TFTs with PVDF-HFP or FPVDF-HFP dielectrics were obtained in the saturation regime with $V_G$ sweep (±5 V) under $V_D$ of 5 V. In contrast to a counter-clockwise hysteresis displayed by the TFT with PVDF-HFP dielectric, no hysteresis was observed for the TFT with FPVDF-HFP dielectric. Due to their high dielectric constant ($k$) values and lack of functional groups that trap charges, PVDF-based polymers are usually employed as dielectrics to fabricate low-voltage-operating electrically stable TFTs[56]. The ferro-electric behavior of these dielectrics (such as PVDF-HFP), which arises

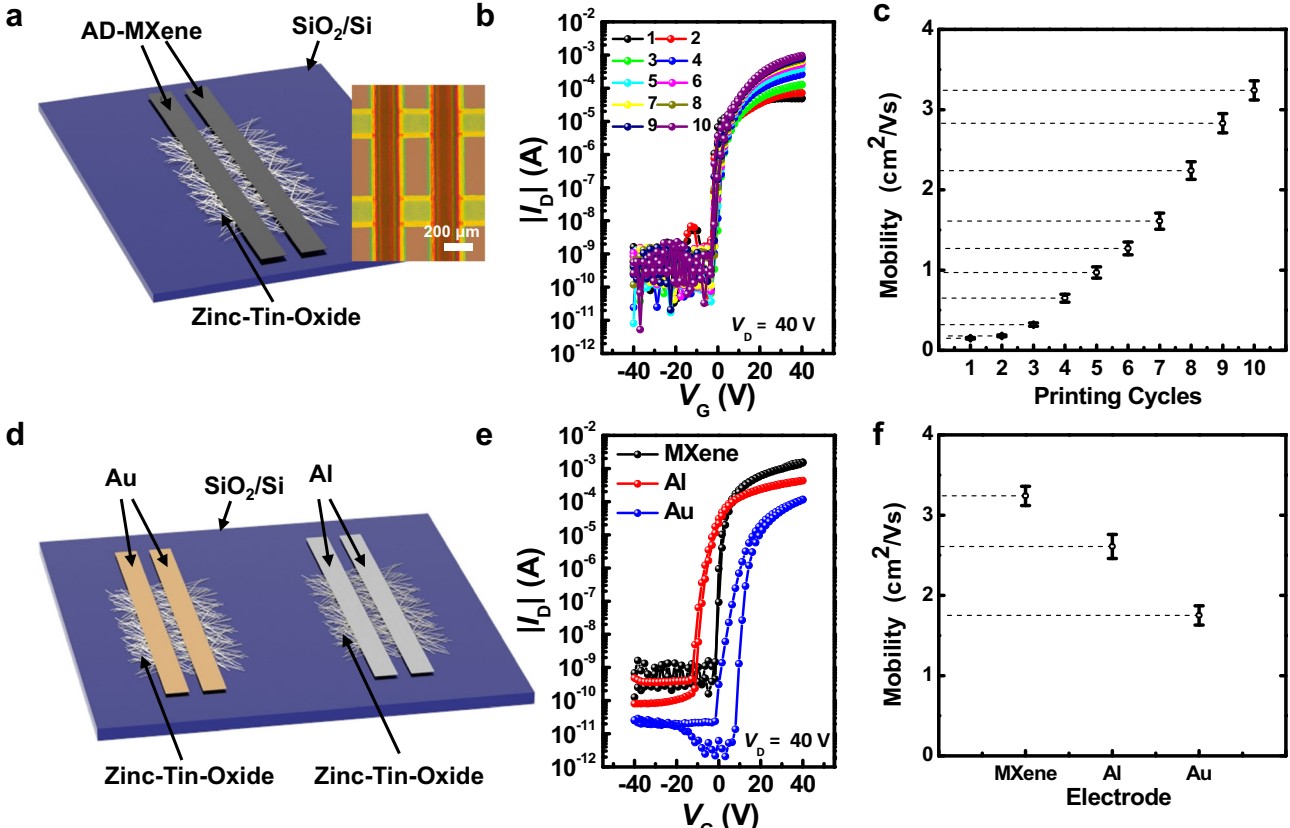

**Fig. 3 | Transfer characteristics and mobilities of TFTs with different electrodes.** **a** Schematic representation and an optical microscope image of top-contact TFT with EHD printed AD-MXene electrodes. Transfer characteristics (**b**) and $\mu_{FET}$ values (**c**) of TFTs employing AD-MXene electrodes produced with various numbers of EHD printing cycles. **d** Schematic representation of top-contact TFTs employing vacuum-deposited Au and Al electrodes. Transfer characteristics (**e**) and $\mu_{FET}$ values (**f**) of TFTs employing AD-MXene, Al, and Au electrodes. The error bars in this figure represent the standard deviations of five different measurements from the same data.

due to the C-F dipoles rearrangement under a strong electric field, results in hysteresis during the operation of TFTs, as shown in Fig. 4b[57]. The size of the hysteresis can be decreased by controlling the crystallinity and crystallite size of the dielectrics, precisely, by decoupling the ferroelectric domains of dielectrics[58]. It was demonstrated that the ferroelectric behavior of the PVDF-HFP dielectric can be substantially decreased by adding the FPA-3F cross-linker[58,59]. Accordingly, negligible hysteresis was observed during the transfer operation of TFT with FPVDF-HFP dielectric layer (Fig. 4c). The electrical parameters of both TFTs, such as $\mu_{FET}$, $V_{th}$, and ON/OFF current ratio ($I_{ON}/I_{OFF}$) were extracted in the saturation regime of the transfer curves and summarized in Table 1. Positive bias-stress tests were performed on the TFTs to confirm the device reliability, where the $V_{th}$ of n-type TFTs shifts towards the positive value under the applied positive gate-source voltage (Fig. 4d, e). The transfer characteristics recorded under the continuous applied bias of 5 V revealed TFTs' good bias stability features. They ascertained the electrical robustness of both PVDF-HFP and FPVDF-HFP dielectrics. Therefore, the fabrication of TFTs with AD-MXene electrodes provides insights into the future manufacturing of MXene-based electronic devices with operational and chemical stability.

The transfer characteristics of the TFT with FPVDF-HFP dielectric were further investigated after one month of storage under 60% RH at 25 °C, as illustrated in Fig. 5a. Transfer curves (Fig. 5b), bias-stress test results (Fig. 5c, d), and measured parameters in Table 1 revealed that the exposure to relative humidity doesn't significantly change the electrical performance of the TFT device. In addition to preserving the bias

**Table 1 | Electrical parameters of TFTs with PVDF-HFP and FPVDF-HFP dielectric layers**

| | $C_i$ (nF cm$^{-2}$) | W/L | $\mu_{FET}$ (cm$^2$V$^{-1}$s$^{-1}$) | $I_{on}/I_{off}$ | $V_{th}$ (V) |
|---|---|---|---|---|---|
| PVDF-HFP | 36.4 | 3.1 | 4.52 ± 0.1 | ~10$^4$ | −0.31 |
| FPVDF-HFP | 41.7 | 3.1 | 4.18 ± 0.1 | ~10$^4$ | −0.03 |
| FPVDF-HFP (One month)$^a$ | 41.7 | 3.1 | 3.46 ± 0.1 | ~10$^4$ | −0.04 |

$^a$Electrical parameters of TFTs with FPVDF-HFP dielectric after the 60% RH exposure at 25 °C for 30 days.

stability feature, the device maintained $\mu_{FET}$, $I_{ON}/I_{OFF}$ ratio, and $V_{th}$ values remained close to their initial values even after the exposure of 60% RH for 30 days. All these results confirmed the excellent stability of AD-MXene electrodes under humid conditions. To demonstrate the scalability of TFTs with AD-MXene electrodes, we fabricated a 7-inch wafer-scale TFT array using FPVDF-HFP dielectric. The digital photograph of the TFT array, along with the optical microscope image of an individual device, is displayed in Fig. 6a. All 64 TFTs in the array exhibited stable transfer curves under $V_G$ sweep from -5 V to 5 V. Furthermore, good output characteristics, i.e., linear/saturation switching with a stepwise increase in the $V_G$, and excellent bias stability were observed for the TFT devices in the array (Supplementary Fig. 8). The $\mu_{FET}$ values calculated from the transfer curves of 64 TFTs were distributed in a small range, with an average $\mu_{FET}$ being 4.42 ± 0.18 cm$^2$ V$^{-1}$ s$^{-1}$ (Fig. 6b).

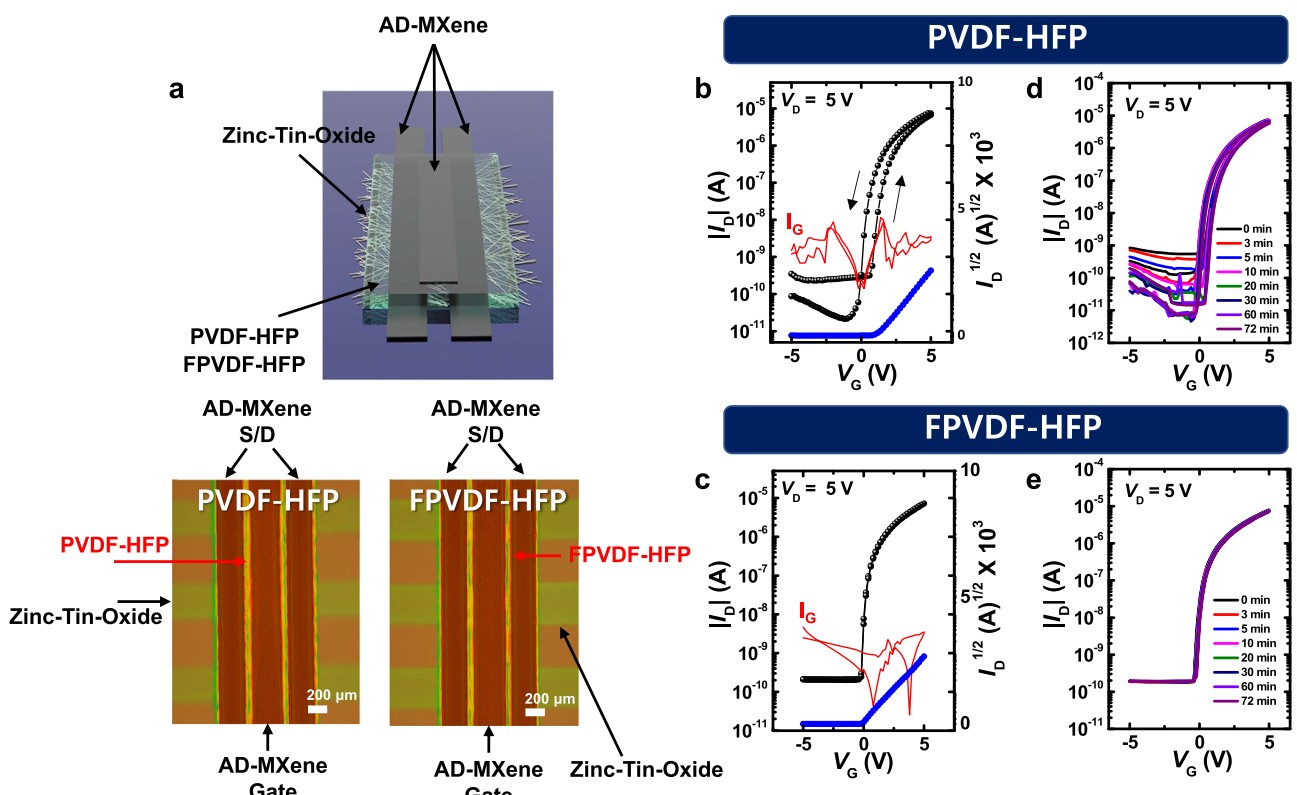

**Fig. 4 | Transfer characteristics and gate bias-stress tests of TFTs with PVDF-HFP or FPVDF-HFP dielectric layers. a** Schematic illustration and top-view optical microscope images of top-gate top-contact TFTs consisting of AD-MXene electrodes, ZTO active layer, and PVDF-HFP or FPVDF-HFP dielectric layer. Transfer characteristics (**b**, **c**) and gate bias-stress tests (**d**, **e**) of the top-gate top-contact TFTs.

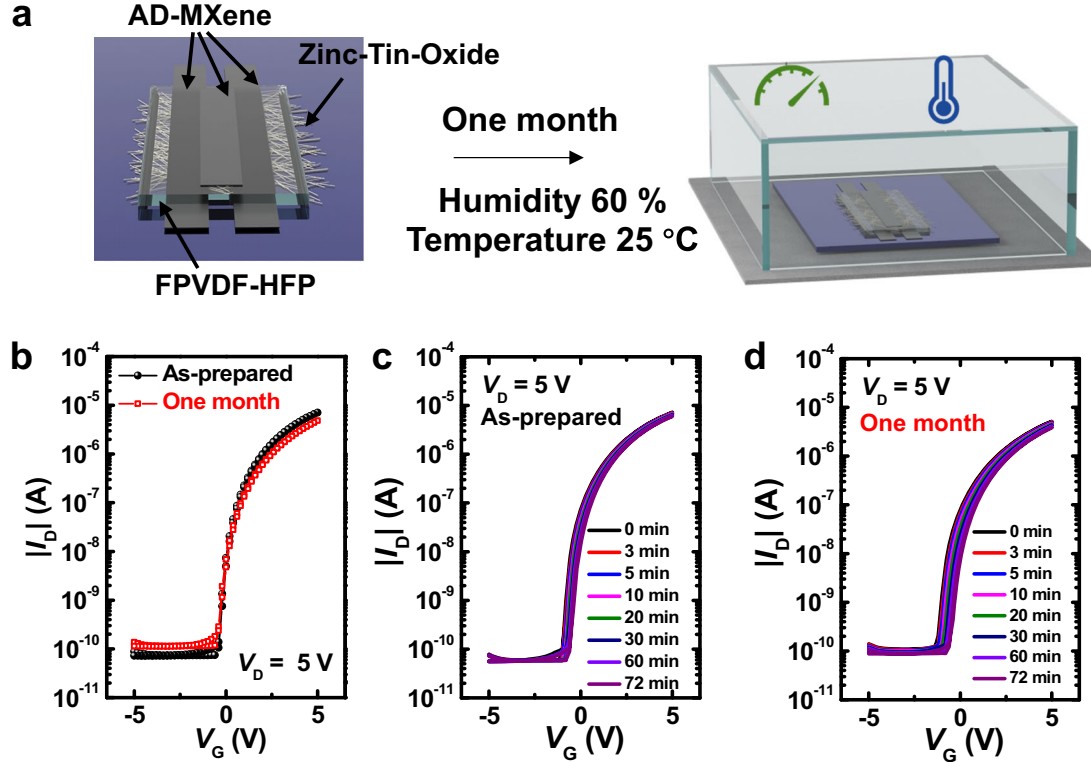

**Fig. 5 | Stability results of TFTs after exposure to humidity (60% RH at 25 °C). a** Schematic showing the endurance test of TFT with AD-MXene electrodes under 60% RH at 25 °C. **b** Transfer characteristics of the TFT before and after 30 days of storage under 60% RH at 25 °C. Bias-stress stability test in the transfer characteristics of the TFT before (**c**) and after 30 days of storage (**d**) under 60% RH at 25 °C.

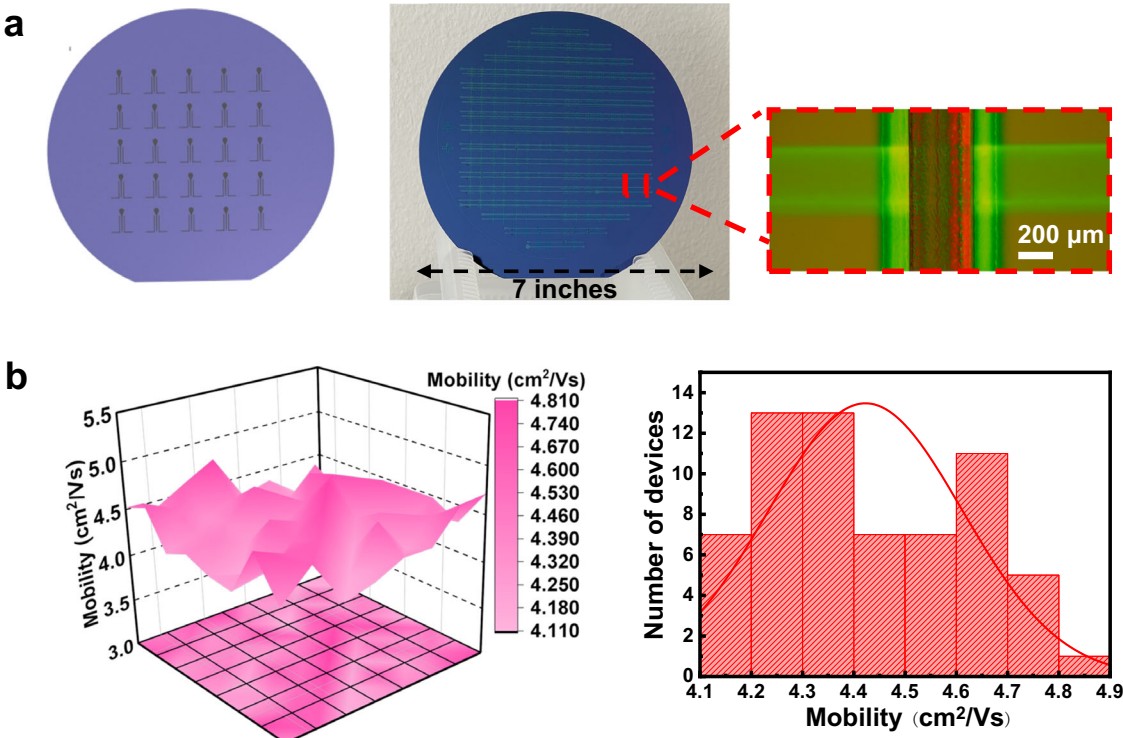

**Fig. 6 | Fabrication of TFT device array and their mobilities. a** Schematic representation and a digital photograph of an array of 64 TFT devices on a 7-inch wafer, and the optical microscope image of an individual TFT device in the array. **b** Scattered plot and histogram showing the distribution of $\mu_{FET}$ values of 64 TFT devices.

## Integration of TFTs for the fabrication of logic circuits

To demonstrate the practical applicability of AD-MXene electrodes, a high-performance complementary inverter was fabricated with TFTs comprising AD-MXene electrodes, FPVDF-HFP dielectric, and ZTO active layer. Figure 7a shows the schematic, top-view optical image, and circuit diagram of the complementary inverter fabricated by using two TFTs (load TFT and drive TFT), wherein the load transistor's gate electrode was connected to the drain electrode, which was used as the power supply voltage ($V_{DD}$). The typical voltage transfer (input–output) characteristics and corresponding inverter gains were investigated at a $V_{DD}$ of 5 V, as shown in Fig. 7b. Before the switching (or at low input voltage ($V_{IN}$)), the inverter's output voltage ($V_{OUT}$) is almost equal to that of $V_{DD}$. However, the high $V_{IN}$ enabled the sharp switching of the inverter's $V_{OUT}$ from a high value to 0 V, and the $V_{OUT}$ was restored to a high value when the $V_{IN}$ decreased back to a low value. This voltage reversal feature of the inverter can transform "1" and "0" input logic states to "0" and "1" output states, respectively. It was also observed that voltage transfer curves of inverters fabricated with FPVDF-HFP dielectric didn't present any hysteresis feature. This inverter exhibited a significantly higher voltage gain ($dV_{OUT}/dV_{IN}$) of 17.8.

The feasibility of prepared TFTs in constructing logic gates was further demonstrated by fabricating NAND and NOR logic gates (Fig. 7c, d). Assembling an additional drive TFT to NOT gate in series and parallel completes the fabrication of NOR and NAND logic gates, respectively. Figure 7c shows top-view optical images and circuit diagrams of NAND and NOR logic gates. The $V_{OUT}$ characteristics of both these logic gates were investigated at low-voltage-operation conditions (-3 V) against different combinations of input signals ($V_A$ and $V_B$) (Fig. 7d). When at least one of the drive TFTs is turned OFF by maintaining 0 V input voltage, i.e., at logic input signal combinations of (0, 0), (1, 0), and (0, 1), the $V_{OUT}$ of the NAND logic gate becomes "$V_{DD}$", i.e., a logic output signal "1". In other words, the $V_{OUT}$ of the NAND logic gate becomes "0 V" or logic state "0" if and only if both drive transistors are turned "ON" by applying $V_A$ = 3 V and $V_B$ = 3 V or the input logic

signals combination (1, 1) (Fig. 7d). On the other hand, the $V_{OUT}$ of NOR gate, wherein the drive TFTs are connected in series, becomes a logic state "1" ($V_{OUT} = V_{DD}$) if and only if both TFTs are turned OFF by maintaining $V_A$ = 0 V and $V_B$ = 0 V or a logic input signal combination (0, 0). Any other combination of input voltages would turn ON at least one of the TFTs. As a result, the $V_{OUT}$ of the NOR gate becomes logic state "0" ($V_{OUT}$ = 0 V) (Fig. 7d).

## Integration of TFTs for the fabrication of 1T1M device

PVDF and its copolymer, PVDF-HFP, are renowned for their ferroelectric behavior, primarily attributed to the reorientation of carbon–fluorine (C–F) dipoles under high electric fields. This trait allows a ferroelectric material to retain its polarization even after removing the electric field. On the other hand, in the case of FPVDF-HFP, which uses fluorophenyl azide to form the cross-links, the reorientation of the atoms was restricted. As a result, removing the electric field eliminated the polarization, allowing for stable driven transistor behavior. The hysteresis observed during the operation of TFT with PVDF-HFP dielectric is identical to that of conventional ferroelectric memory transistors (Fig. 4b). Hence, the feasibility of these TFTs for integrating memory devices was investigated. The transfer characteristics of TFT with PVDF-HFP dielectric were shifted depending on the applied constant positive and negative $V_G$, as shown in Fig. 8a. In detail, shifting towards negative values was noted when increasing the constant positive $V_G$ (5 V, 10 V, and 20 V), while shifting towards positive values was noted when increasing the constant negative $V_G$ (-5 V, -10 V, and -20 V). Accordingly, the absolute $V_{th}$ value shift towards positive and negative values increased with an increase in the constant negative and constant positive $V_G$, respectively (Fig. 8b). Different $I_D$ values observed at $V_G$ = 0 V for constant positive and negative $V_G$ can act as ON- and OFF-states of the device, and are helpful for the information programming or erasing. The ON-state current of $\approx 10^{-6}$ A was measured after applying a constant positive voltage of 20 V, while the off-state current of $\approx 10^{-10}$ A was measured after applying a constant negative voltage of −20 V (Fig. 8c, d). The ON-state current allows the

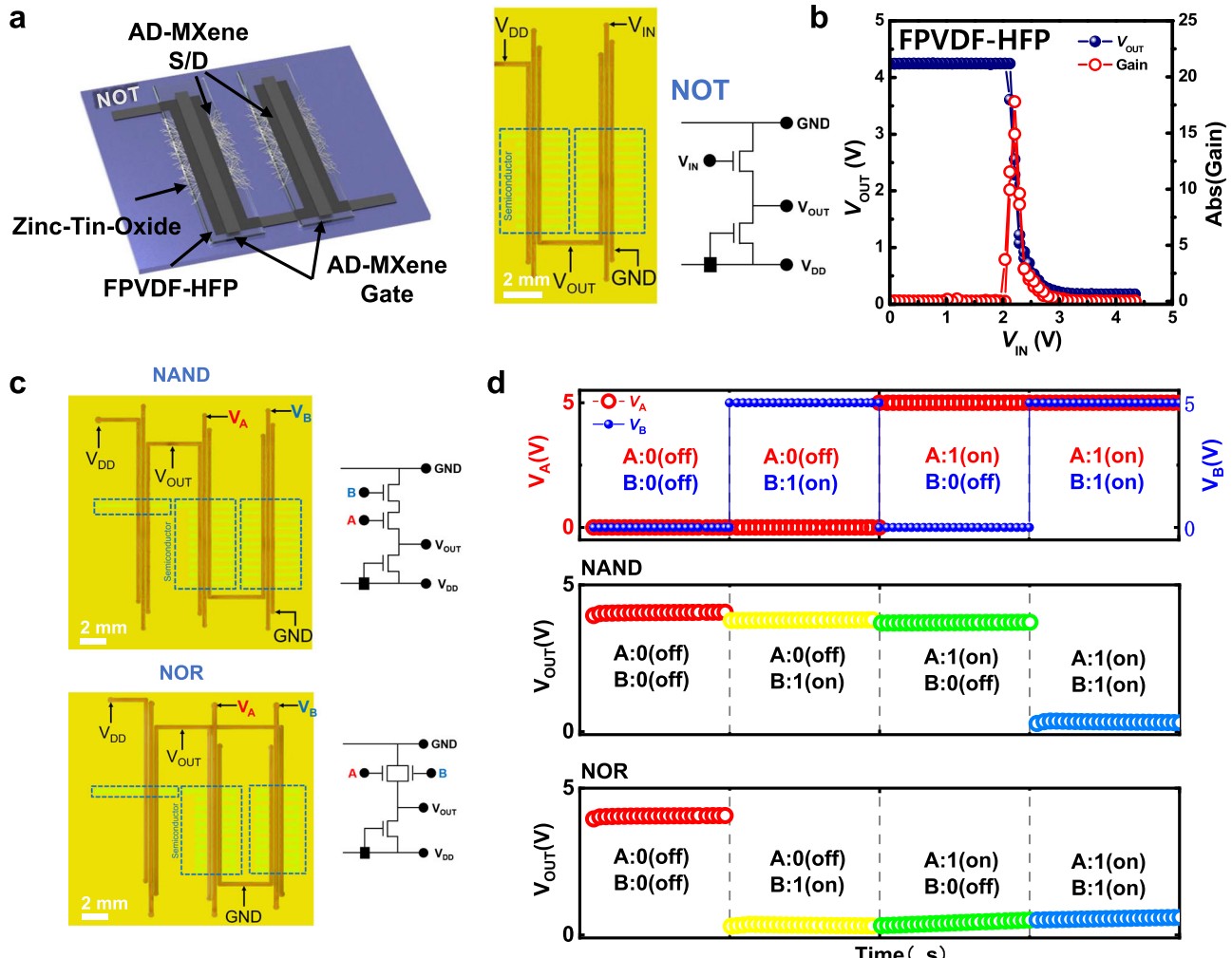

**Fig. 7 | Output characteristics of NOT, NAND, and NOR logic gates. a** Schematic representation, optical microscope image, and circuit diagram of the fabricated complementary inverter device. **b** Voltage transfer characteristics and DC voltage gain of the complementary inverter fabricated with AD-MXene electrodes and FPVDF-HFP dielectrics. **c** Optical microscope images and circuit diagrams of the developed NAND and NOR logic gates. **d** Input voltages and corresponding output voltage characteristics of NAND and NOR logic gates fabricated with AD-MXene electrodes and FPVDF-HFP dielectric.

device to be programmed for information storage, while the erasing can be achieved with the OFF-state current. Further, upon increasing the erasing pulse width from 0.3 to 2 s, the transfer curve shifted in the positive $V_G$ direction. However, this tuning of pulse width didn't result in any variation corresponding to $I_D$ at $V_G = 0$ (Supplementary Fig. S9).

The prepared TFTs with PVDF-HFP and FPVDF-HFP dielectrics were used as memory TFT and control TFT, respectively, for fabricating one-transistor-one-memory (1T1M) cells. The 1T1M architecture was chosen to ensure a non-destructive read-out capability for ferroelectric memory cells. This design offers the advantage of separating the distinct programming/erasing and reading processes of the memory transistor by utilizing the control transistor. Figure 9a shows the schematic representation of the fabricated memory device along with an optical micrograph and circuit diagram. The gate voltage of the memory TFT can be applied depending on the signal of the control TFT. The dynamic behavior of the 1T1M device was examined by implementing multiple writing and reading processes, as shown in Fig. 9b. Programming/erasing and reading operations were distinctively selected by controlling bit line voltage ($V_{BL}$) signal access to the memory transistor through applying word line voltages ($V_{WL}$) of 20 V and −20 V, respectively. In detail, the control transistor was turned ON upon applying a $V_{WL}$ of 20 V; subsequently, this action triggered the transfer of the $V_{BL}$ signal to the gate electrode of the

memory transistor. In this condition, both programming and erasing operations were accomplished by applying $V_{BL}$ as 20 V and −20 V, respectively. In contrast, the reading of the stored data was performed at $V_{WL} = -20$ V, wherein the control transistor was turned OFF, and, as a result, the $I_D$ of the memory transistor remained at the same state irrespective of the $V_{BL}$. This fabricated 1T1M cell exhibited good retention and cyclic endurance properties, as shown in Fig. 9c, d, respectively. After adjusting the memory cell to a programming or erasing state, the retention test was performed by monitoring the readout current as a function of time at $V_{WL} = -20$ V. The device maintained ON and OFF currents for $10^4$ s without much degradation in performance, which specified its high retention capability. The cycling stability test, which was conducted by recording the currents for repeated cycles of programming and erasing processes, revealed the switching stability and reliability of the device (Fig. 9d).

In summary, we demonstrated the patterning of alkylated 3,4-dihydroxy-L-phenylalanine (ADOPA) ligand functionalized MXene (AD-MXene) dispersions in ethanol by electrohydrodynamic (EHD) printing process. The excellent conductivity and facile processability of AD-MXene lines allowed their utilization as source, drain, and gate electrodes to fabricate all-solution processed thin-film transistors (TFTs) through EHD printing, wherein zinc-tin-oxide and PVDF-HFP or FPVDF-HFP were utilized as active and dielectric layers of TFTs, respectively.

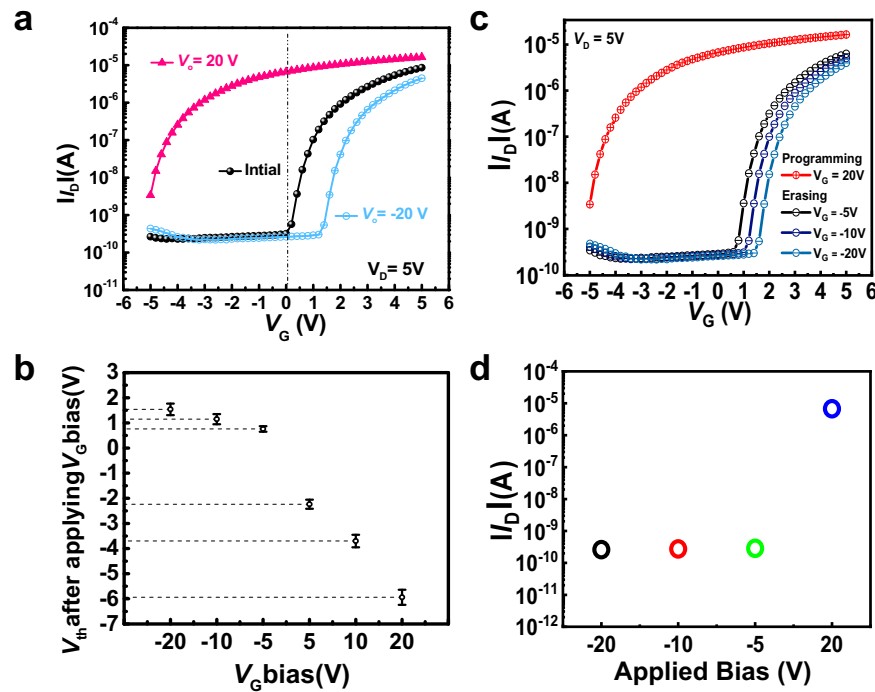

**Fig. 8 | Transfer characteristics of TFTs under the application of programming and erasing levels. a** Transfer characteristics of TFT with AD-MXene electrodes and PVDF-HFP dielectric after applying a negative gate bias (−20 V) for erasing levels and positive gate bias (20 V) for programming levels. **b** $V_{th}$ of the TFT after applying different programming and erasing levels. **c** TFT transfer curves after applying programming (20 V) and erasing levels (−5, −10, and −20 V) for 1 s. **d** $I_D$ values of the TFT at $V_G = 0$ after applying programming (20 V) and erasing levels (−5, −10, and −20 V). The error bars in this figure represent the standard deviations of five parallel tests.

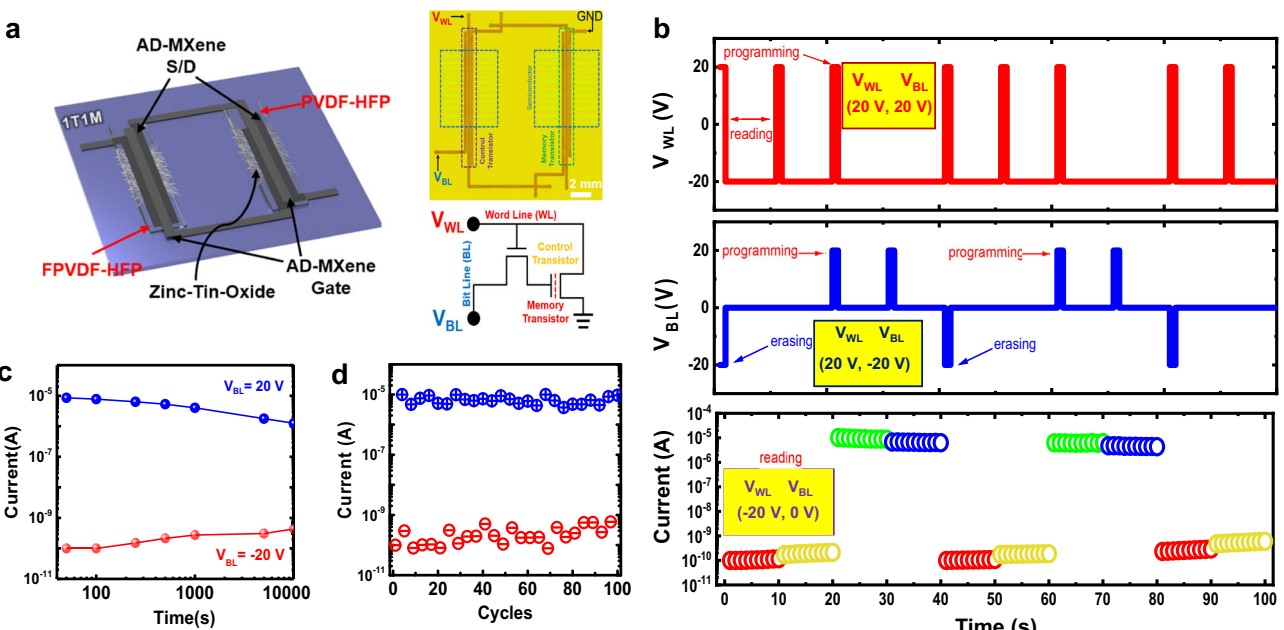

**Fig. 9 | Dynamic response, retention, and cyclic stability properties of 1T1M cell. a** Schematic illustration, optical microscope image, and circuit diagram of the fabricated 1T1M cell. **b** Dynamic response of the 1T1M cell, i.e., $I_D$ as a function of $V_{BL}$ and $V_{WL}$ input signals: $V_{WL} = 20$ V and $W_{BL} = 20$ V for programming, $V_{WL} = 20$ V and $W_{BL} = -20$ V for erasing, and $V_{WL} = -20$ V and $W_{BL} = 0$ V for reading. Retention properties (**c**) and cyclic stability over 100 cycles (**d**) of programming/erasing operations of the 1T1M cell.

The performance of AD-MXene electrodes was higher than that of the vacuum-deposited Au and Al contacts. We demonstrated exceptional stability to TFTs under high humidity conditions due to hydrophobic ADOPA ligands' ability to restrict moisture ingress. Furthermore, the robust operation of TFTs with AD-MXene electrodes facilitated the fabrications of high-performance complementary logic circuits (complementary inverter, NAND gate, and NOR gate) and a one-transistor-one-memory cell. Overall, this work proves that appropriately designed surface functionalized MXenes provide a pathway to environmentally stable printable MXene-based electronics (MXetronics).

## Methods

### ADOPA-MXene ink preparation

The syntheses of ADOPA ligand, MXene, and AD-MXene were carried out following our earlier report[39].

**Synthesis of ADOPA ligand.** ADOPA ligand was synthesized by esterification of 3,4-dihydroxy-L-phenylalanine (DOPA) with hydrophobic $1H,1H,2H,2H$-perfluoro-1-hexanol. Briefly, 0.2 M toluene (100 mL) solution containing 20 mmol of DOPA (3.94 g), 22 mmol of $1H,1H,2H,2H$-perfluoro-1-hexanol (5.81 g), and 20 mmol of $p$-TsOH·$H_2O$ (3.80 g) was refluxed for 48 h under an Ar atmosphere using a Dean-Stark trap to remove water azeotropically. After the completion of the reaction, toluene was evaporated under vacuum to get the gel-like solid residue, which was thoroughly washed with saturated $NaHCO_3$ and subsequently extracted with ethyl acetate. After that, the obtained ethyl acetate solution was washed with brine and dried over anhydrous $MgSO_4$ to remove water. The resulting residue was dissolved in a small amount of hot ethyl acetate and crystallized from petroleum ether (60–90 °C) to produce ADOPA (6.56 g, 74%).

**Synthesis of $Ti_3C_2T_x$ MXene.** The selective removal of Al layers from the $Ti_3AlC_2$ MAX precursor using the modified minimally intensive layer delamination (MILD) method facilitated the exfoliation of $Ti_3C_2T_x$ MXene nanosheets. Briefly, 3 g of $Ti_3AlC_2$ MAX powder was stirred in 60 mL of 9 M HCl aqueous solution comprising 4.8 g of LiF for 24 h at 35 °C. The crude product obtained was washed several times via centrifugation (1246 × g for 5 min) until the supernatant's pH reached ~6. Finally, delaminated $Ti_3C_2T_x$ nanosheets were obtained by collecting the supernatant solution from the $Ti_3C_2T_x$ dispersions subjected to centrifugation at 1246 × g for 30 min.

**Preparation of AD-MXene ink.** AD-MXene was synthesized by adding 35 mL of the as-prepared aqueous MXene dispersions (1 mg mL$^{-1}$) into 10 mL of ethanol solution containing 3.5 mg of ADOPA under continuous stirring at room temperature. After a brief reaction, the solvent was exchanged with ethanol via centrifugation (10174 × g, 3 times) to obtain the AD-MXene dispersions in ethanol. The AD-MXene dispersions in ethanol were diluted to 5 mg mL$^{-1}$ for further use.

### Fabrication of TFTs

The TFT devices were fabricated on a heavily n-doped Si wafer with a 100 nm-thick thermally grown $SiO_2$ layer. EHD printing was used to sequentially print active layers, source/drain electrodes, dielectrics, and gate electrodes to form TFTs with top-gate top-contact architecture. At first, $SiO_2$/Si substrates were cleaned by immersing in boiling acetone, followed by sonication in acetone and isopropyl alcohol for 30 min each, dried under nitrogen gas, and further exposed to UV-ozone for 30 min to eliminate organic residues. The ZTO precursor solution was prepared by dissolving 0.4 M zinc acetate dehydrate and tin chloride (1:1 ratio) in 2-methoxyethanol. The $SiO_2$/Si substrates were patterned with the ZTO active layer using EHD printing, and then the resulting film was annealed at 500 °C for 2 h to form the ZTO active layer. AD-MXene source and drain electrodes were patterned by printing AD-MXene ink via the cone-jet mode EHD printing process. The residual solvent from AD-MXene electrodes was removed through heat treatment at 120 °C. Afterward, either 7 wt% PVDF-HFP or FPVDF-HFP (a solution with a weight concentration of 7 wt% was prepared by dissolving PVDF-HFP and fluorophenyl azide (FPA-3F) in dimethylacetamide at a weight ratio of 95:5.) dielectric solutions were printed in direct contact mode (electrostatic-force-assisted dispensing), wherein the DC bias, flow rate, and working distance were 0.05 kV, 0.8 μL min$^{-1}$, and 80 μm, respectively. The deposited dielectric layers were thermally annealed at 120 °C for 1 h. The film prepared (FPVDF-HFP) was subsequently cured by exposing it to UV light at a wavelength of

256 nm in the presence of $N_2$ gas to solidify the dielectric layer. Finally, the EHD printing of the ADOPA-MXene gate electrode completed the device fabrication. For comparison, TFTs with conventional metal electrodes were fabricated by employing the same fabrication conditions except for replacing AD-MXene electrodes printing step with the vacuum-deposition of Au or Al electrodes with the thickness being the same as AD-MXene electrodes (190 nm). Channel length and width of the ZTO layer were 236 μm and 731 μm, respectively.

### Characterizations

X-ray photoelectron spectroscopy measurements were performed on the ULVAC-PHI XPS instrument (Japan) with Al Kα radiation (1486.6 eV). X-ray diffraction (XRD) patterns of pristine MXene and AD-MXene were acquired using a D8 diffractometer (Bruker, USA) with a Cu K$_\alpha$ radiation source. UV-visible absorbance spectra were recorded on a V-670 instrument (Jasco, Japan). The AD-MXene flakes morphology and their arrangement in printed AD-MXene electrodes were examined using a field emission scanning electron microscope (JEOL, JSM-7610F). In contrast, the morphologies of printed AD-MXene electrodes and devices were studied using an optical microscope (Nikon ECLIPSE LV100ND). The dielectric performance was evaluated using an LCR meter and a Keithley 4200 SCS. The electrical characteristics of the TFTs and integrated devices (logic gates and 1T1M cells) were measured under vacuum (-10$^{-3}$ Torr) using a Keithley 4200 SCS unit. The surface morphology and roughness of AD-MXene lines were measured by atomic force microscope (XE-100, Park systems) in non-contact mode.

### Reporting summary

Further information on research design is available in the Nature Portfolio Reporting Summary linked to this article.

## Data availability

The data supporting this study's findings are available within the article and Supplementary Information. Source data are provided with this paper.

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

## Acknowledgements

This research was supported by the National Research Foundation of Korea (NRF) grant funded by the Korea government (MSIT): 2022M3J7A1062940 (S.J.P. and I.I.), 2022R1F1A1064314 (Y.J.J.), 2023R1A2C1004109 (S.J.P.), and 2020M3H4A3081819 (S.H.K.). This research was also supported by the Basic Science Research Program through the National Research Foundation of Korea (NRF), funded by the Ministry of Education: 2018R1A6A1A03023788 (I.I.) and 2021R1I1A1A01055790 (G.M.). This work was supported by the National Research Council of Science & Technology (NST) grant from the Korea government: CRC22031-000 (S.J.K.). This research was supported by Korea Electric Power Corporation: R21XO01-5 (S.J.P. and I.I.). This work was supported by the Leaders in Industry-University Cooperation Phase 3 (LINC 3.0) project funded by the South Korean Ministry of Education (S.J.P. and I.I.). T.Y.K., G.M., Y.H.P., and I.I. thank Nanoplexus Ltd, UK, for their support and valuable discussions.

## Author contributions

T.Y.K., H.Y., G.M., S.J.K., S.H.K., S.Y.L., and I.I. conceived the main ideas. T.Y.K., H.Y., and G.M. performed the overall experiments and characterizations. Y.H.P., J.Y.L., J.H.L., and D.J.Y. contributed to the characterization and discussed the overall experimental concepts. T.Y.K., G.M., Y.J.J., and I.I. drafted the manuscript. S.J.K., S.H.K., S.J.P., and Y.G. discussed the data, reviewed and edited the manuscript. I.I. supervised this study.

## Competing interests

The authors declare no competing interests.

## Additional information

[1]Materials Architecturing Research Center, Korea Institute of Science and Technology, 5, Hwarang-ro 14-gil, Seongbuk-gu, Seoul 02792, South Korea. [2]Convergence Research Center for Solutions to Electromagnetic Interference in Future-mobility, Korea Institute of Science and Technology, 5, Hwarang-ro 14-gil, Seongbuk-gu, Seoul 02792, South Korea. [3]Nanoplexus Solutions Ltd, Graphene Engineering Innovation Centre, Masdar Building, Sackville Street, Manchester M1 3BB, UK. [4]School of Flexible Electronics (SoFE) and Henan Institute of Flexible Electronics (HIFE), Henan University, 379 Mingli Road, Zhengzhou 450046, China. [5]School of Chemical Engineering, Konkuk University, Seoul 05029, South Korea. [6]Department of Polymer Science and Engineering, Chemical Industry Institute, Korea National University of Transportation, Chungju 27469, South Korea. [7]Department of IT-Energy Convergence (BK21 FOUR), Korea National University of Transportation, Chungju 27469, South Korea. [8]Department of Chemistry, Inha University, Inharo 100, Incheon 22212, South Korea. [9]Department of Materials Science and Engineering, Korea University, 145, Anam-ro, Seongbuk-gu, Seoul 02841, South Korea. [10]Analytical Science Laboratory of Samsung Advanced Institute of Technology (SAIT), Suwon 16678, South Korea. [11]Department of Materials Science and Engineering and A. J. Drexel Nanomaterials Institute, Drexel University, Philadelphia, Pennsylvania 19104, US. [12]Division of Nanoscience and Technology, KIST School, University of Science and Technology, 5, Hwarang-ro 14-gil, Seongbuk-gu, Seoul 02792, South Korea. [13]Department of Materials Science and Engineering, Korea National University of Transportation, Chungju 27469, South Korea. [14]These authors contributed equally: Tae Yun Ko, Heqing Ye, G. Murali, Seul-Yi Lee. ✉e-mail: Seonjkim@kist.re.kr; shkim97@konkuk.ac.kr; yjjeong@ut.ac.kr; sjpark@inha.ac.kr; in1@ut.ac.kr

