## [Peer Review File · Nature Communications]

REVIEWER COMMENTS

Reviewer #1 (Remarks to the Author):

The manuscript "Functionalized MXene Ink Enables Environmentally Stable Printed Electronics" by Ko et al., reported on a clever strategy to prepare MXene inks to facilitate printing techniques for integrated circuit fabrication and improve electrical performance. They introduced a functional ligand with catechol-based head group and hydrophobic tails, ADOPA, for the surface treatment of solution-processed MXene nanosheets. While the electrical conductivity of MXene remains high, successful surface functionalization of MXene nanosheets with ADOPA ligands produced the well-dispersed ink in ethanol solvent. As a result, the electrodes of electronic devices were fabricated using the EHD printing method with highly dispersed MXene ink and demonstrated to exhibit excellent device performance comparable to deposited metal electrodes. Because the paper is interesting and the subject is suitable for 'Nature Communication', I recommend a minor revision of this manuscript.

Comment 1: The author used PVDF-HFP and FPVDF-HFP dielectrics for the fabrication of 1T1M device. It would be better if the author performs in-depth study on the difference between PVDF-HFP and FPVDF-HFP dielectrics in the printed MXene TFTs.

Comment 2: The author should mention what concentration ADOPA-functionalized MXene stably disperses in ethanol.

Comment 3: Compared to EHD printing, ink jet printing technology is expected to be more universal and applicable to real industries. The author should mention in the introduction why you chose EHD printing instead of ink-jet printing in this study. Also, it would be better that this work includes an ink-jet printing study of the developed functionalized MXene ink.

Reviewer #2 (Remarks to the Author):

The paper outlines a novel fabrication process for environmentally stable thin-film transistors (TFTs) using MXene by electrohydrodynamic (EHD) printing. The study is innovative and contributes to the field of printed transistors. However, I believe that the paper could be significantly improved by addressing the following points:

1. The manuscript needs a clearer statement of the significance of the work in the context of existing research. Additionally, the challenges addressed by the proposed fabrication process should be highlighted more explicitly. A thorough literature review should be conducted to better situate the work within the current state-of-the-art and to articulate the motivations driving this research.

2. Between Lines 148-157, the manuscript would benefit from a more detailed description of the EHD printing parameters, including nozzle size, printing speed, the presence of any back pressure during printing, and other relevant operational conditions.
3. While the technical aspects of the fabrication process are well-discussed, the manuscript lacks in-depth scientific analysis. Please provide data on the conductivity of the printed MXene for different layer thicknesses, as well as surface roughness measurements. Furthermore, a detailed explanation regarding why MXene electrodes exhibit higher mobility compared to Al and Au electrodes would strengthen the scientific foundation of the paper.
4. In Lines 214-225, the manuscript reports MXene TFTs are with high mobility as well as the higher off current, but does not clarify whether this characteristic is beneficial or detrimental. Please provide a more detailed discussion on that.
5. There appears to be a font inconsistency at line 214. Please ensure that the font type and size are consistent throughout the manuscript for a professional and polished presentation.
6. Please provide a justification for selecting PVDF-HP and FPVDF-HP as the dielectric layer. Discuss its advantages or properties that make it suitable for this application compared to other materials.
7. In the Performance of AD-MXene as electrodes of TFTs, please include the channel length and width to aid in understanding the device configuration.
8. It could be better to add the optical microscopy image of AD-MXene as electrodes of TFTs in Fig. 3.
9. In Fig. 4, label each component in the microscopy images to aid readers in interpreting the results.
10. The manuscript demonstrates the stability of the TFTs from FPVDF-HP but does not show how MXene contributes to this stability. Please provide evidence or a discussion that supports the role of MXene in the stability of the TFTs which makes it align with the title.
11. A scale bar should be inserted in Fig. 6, Fig. 7 and Fig. 9 to provide a reference for size and dimension.

In conclusion, I recommend that the paper be accepted after the authors have made the major revisions outlined above. The revisions will significantly improve the manuscript and ensure that the scientific contributions are clearly communicated.

REVIEWER COMMENTS

Reviewer #1 (Remarks to the Author)

The manuscript "Functionalized MXene Ink Enables Environmentally Stable Printed Electronics" by Ko et al., reported on a clever strategy to prepare MXene inks to facilitate printing techniques for integrated circuit fabrication and improve electrical performance. They introduced a functional ligand with catechol-based head group and hydrophobic tails, ADOPA, for the surface treatment of solution-processed MXene nanosheets. While the electrical conductivity of MXene remains high, successful surface functionalization of MXene nanosheets with ADOPA ligands produced the well-dispersed ink in ethanol solvent. As a result, the electrodes of electronic devices were fabricated using the EHD printing method with highly dispersed MXene ink and demonstrated to exhibit excellent device performance comparable to deposited metal electrodes. Because the paper is interesting and the subject is suitable for 'Nature Communication', I recommend a minor revision of this manuscript.

Comment 1: The author used PVDF-HFP and FPVDF-HFP dielectrics for the fabrication of 1T1M device. It would be better if the author performs in-depth study on the difference between PVDF-HFP and FPVDF-HFP dielectrics in the printed MXene TFTs.

RESPONSE: Thanks for the reviewer's valuable comment. The PVDF-HFP and FPVDF-HFP used in this study have been shown to show excellent stability in 1T1M devices as dielectric materials, making them suitable for TFT research applications. (*ACS Appl. Mater. Interfaces* 2021, 13,11043.). For potential readers of the manuscript, we have addressed this information by mentioning it in the "**Results and Discussion**" section of the manuscript, made corresponding modifications.

The modified content can be found on page 16, line 378-384

Integration of TFTs for the fabrication of 1T1M device. PVDF and its copolymer, PVDF-HFP are renowned for their ferroelectric behavior, primarily attributed to the reorientation of carbon-fluorine (C-F) dipoles under high electric fields. This unique trait allows a ferroelectric material to retain its polarization even after the removal of the electric field. On the other hand, in the case of FPVDF-HFP, which uses fluorophenyl azide to form the

cross-links, the reorientation of the atoms was restricted. As a result, removing the electric field eliminated the polarization, allowing for stable driven transistor behavior.

The modified content can be found on page 18, line 412-415

The prepared TFTs with PVDF-HFP and FPVDF-HFP dielectrics were used as memory TFT and control TFT, respectively, for fabricating 1TIM cells. The one-transistor-one-memory (1TIM) architecture was chosen to ensure a non-destructive read-out capability for ferroelectric memory cells. This design offers the advantage of separating the distinct programming/erasing and reading processes of the memory transistor (MT) by utilizing the control transistor (CT). Fig. 9a shows the schematic representation along with an optical micrograph and circuit diagrams of the fabricated memory device.

Comment 2: The author should mention what concentration ADOPA-functionalized MXene stably disperses in ethanol.

RESPONSE: Thanks for the reviewer's valuable comment. A uniform dispersion of AD-MXene in ethanol can be achieved up to as high as 50 mg mL⁻¹ concentration, which was demonstrated in our previous study (ACS Nano 2023, 17, 2, 1112–1119). For the present investigation, we have used diluted AD-MXene dispersions in ethanol with a concentration of 5 mg mL⁻¹ for successful jetting during the EHD printing, and it was mentioned in "Preparation of AD-MXene ink" subsection of the Experimental Section.

Comment 3: Compared to EHD printing, ink jet printing technology is expected to be more universal and applicable to real industries. The author should mention in the introduction why you chose EHD printing instead of ink-jet printing in this study. Also, it would be better that this work includes an ink-jet printing study of the developed functionalized MXene ink.

RESPONSE: We thank the reviewer for valuable comment. Both inkjet printing and EHD (electrohydrodynamic) printing are digital printing methods, and each has its advantages. The developed AD-MXene flakes for the present study have a large size (~ 2 μm), making it challenging to achieve uniform printing using the narrow nozzles that an inkjet printer commonly required. Additionally, while AD-MXene flakes disperse well in ethanol, ethanol is not a suitable solvent for the inkjet printing. Therefore, for this study, we opted for the

EHD printing method, which operates well with the low-surface tension ethanol solvent. Unlike the inkjet printing, EHD printing involves the ejection of the solution by an external electric field. Since the ejection is driven by the electric field, the frequency of nozzle clogging is much lower compared to inkjet printing. It is suitable for continuous line printing and similar applications. **Collectively, when choosing a digital printing method in the laboratory, EHD has an advantage over inkjet due to these factors.** Dragging a nozzle during EHD cone-jet mode printing involves shear forces that can induce MXene flakes alignment along the printing direction. This alignment is in accordance with the electron carrier transport direction, thereby leading to lower resistance of AD-MXene electrodes.

As suggested by the reviewer, our research group also wanted to explore printing MXene using the inkjet printing method.

- We attempted to use 2-methoxy ethanol as a solvent, more suitable for inkjet printing than ethanol. However, using 2-methoxy ethanol as the solvent did not result in effective dispersion of AD-MXene flakes.
- We attempted surface modification of MXene flakes using a different ligand for inkjet printing, and in this case, the printing was successful. However, we have also used the mixture of AD-MXene dispersions in ethanol and different ligand functionalized MXene dispersions in 2-methoxy ethanol to demonstrate the ink-jet printing. Although the ink-jet printing was successful in this case (**Figure R1**), due to the frequent nozzle clogging, low-concentration of flakes to ensure good dispersion, and poor conductivity of the printed lines (**Figure R2**) making it difficult for employing the electrode's preparation. Furthermore, the line edge is not clear in the ink-jet printed line pattern due to the coffee-ring effect (**Figure R1**).

Hence, it is not suitable for source and drain applications of transistors where charge injection and extraction efficiency are important.

Figure R1. Optical microscope images of ink-jet printed lines using the mixtures of AD-MXene in ethanol and different ligand functionalized MXene in 2-methoxy ethanol.

Figure R2. I-V characteristics of the ink-jet printed lines using the mixtures of AD-MXene in ethanol and different ligand functionalized MXene in 2-methoxy ethanol.

Reviewer #2 (Remarks to the Author)

The paper outlines a novel fabrication process for environmentally stable thin-film transistors (TFTs) using MXene by electrohydrodynamic (EHD) printing. The study is innovative and contributes to the field of printed transistors. However, I believe that the paper could be significantly improved by addressing the following points:

Comment-1: The manuscript needs a clearer statement of the significance of the work in the context of existing research. Additionally, the challenges addressed by the proposed fabrication process should be highlighted more explicitly. A thorough literature review should be conducted to better situate the work within the current state-of-the-art and to articulate the motivations driving this research.

RESPONSE: We thank the reviewer for valuable comment. This work aimed at addressing the challenges that are associated with MXenes utilization as electrical contacts; and EHD printing of MXenes.

Solution processible 2D materials as electrical contacts present several crucial advantages over conventional vacuum-deposited metal electrodes, such as the ease of processing (spin/spray coating, printing, etc.), avoid the complex fabrication conditions, large-area fabrication, scalability, cost-effectiveness, and compatibility with flexible substrates (*Nat Rev Mater* **8**, 651–667 (2023)). In addition, the sputtering techniques that are used for the deposition of conventional metal electrodes could damage or dope the underneath sensitive (semiconductor or dielectric) material layer owing to the bombardment of sputter ions, making the interfaces at the electrical contacts non-ideal that reduce the device performance (*Matter* **4**, 3549–3584, 2021). Hence, establishing solution processible electrical contacts with excellent efficiency is significantly important for modern electronics.

Titanium carbide MXene ($\text{Ti}_3\text{C}_2\text{T}_x$), exhibiting highest electrical conductivity among all solution processible 2D materials, has been regarded as a prospective material for electrical contacts. But, the use of MXenes in electronics is limited by the gradual degradation of MXene film's electrical conductivity under atmospheric conditions. The hydrophilicity of

MXene's surface functional groups and Li^+ intercalants, originated from the MXene synthesis, allows the penetration of atmospheric water molecules through architectural defects of MXene film. The intercalation of water molecules causes the swelling and oxidation of MXene film/flakes and, as a result, leads to the gradual degradation of MXene film's electrical conductivity. To overcome this, a combination of additional processing strategies for the deposited MXene thin films such as thermal annealing and subsequent surface treatment with hydrophobic barrier layers were demonstrated to be required (*Adv. Mater.* 2022, 34, 2206377). However, these multiple strategies may not suit for all electronic device fabrication conditions/substrates, increase the device fabrication cost, and time-consuming. On the other hand, the poor interaction of pristine MXene nanosheets with the substrate could potentially result in the accumulation of MXene nanosheets at the edges of the droplets (coffee-ring effect) during the printing and subsequent solvent evaporation (*Nature Nanotech* 12, 343–350 (2017)). An additional binder molecule should be mixed with the MXene ink to avoid the coffee-ring effects or to improve the adhesion of MXene films (*Adv. Funct. Mater.* 2018, 28, 1801972). But the mixing of organic molecules/polymers could potentially decline the overall electrical conductivity of resultant MXene film through impeding the efficient inter-flake electron transport in MXene film (*Natl. Acad. Sci. U.S.A.* (2014) 111 (47), 16676; *ACS Nano* (2019) 13 (12), 13818; *Cell Reports Physical Science* (2020) 1 (4), 100042). Therefore, the current available MXene ink formulations and strategies for creating electrical contacts with excellent oxidation stability are still far from ideal.

On the other hand, the low surface tension solvent selectivity of the EHD printing technique restricts the printing of pristine MXene that shows excellent dispersion only in high surface tension water solvent. Therefore, for enabling the EHD printing of MXenes, it is imperative to develop the stable dispersions of MXene nanosheets in low surface tension solvent such as ethanol. Further, low-boiling point solvents such as ethanol are good for printing as they require low-drying temperatures to remove the solvent from printed electrical lines.

Benefits of AD-MXene: The developed AD-MXene addresses above mentioned challenges because of its excellent properties including hydrophobic surface that enable excellent oxidation stability under harsh environmental conditions, high electrical conductivity, excellent adhesion on various substrates, ability to form compact films, and feasible to EHD

printing owing to excellent dispersion stability in ethanol (ACS Nano 2023, 17, 2, 1112–1119).

To effectively highlight the significance of the work for potential readers benefit, **we have added following sentences in the introduction section of the revised manuscript.**

The added content can be found **in page 3, line 55-58**

“Solution processible two-dimensional (2D) materials as electrical contacts offer several crucial advantages over conventional vacuum-deposited metals such as the ease of processing, avoid the complex fabrication conditions, large-area fabrication, scalability, cost-effectiveness, compatibility with flexible substrates, and non-damaging to the underneath surface.^{1, 2}”

The added content can be found **in page 5, line 116-119**

“In contrast to the existing literature, the AD-MXene electrodes doesn't require the complex post-fabrication strategies for avoiding the environmental degradation⁴⁹; and the mixing of binder molecules to the ink, which is detrimental to the electrode conductivity, for enhancing the electrodes adhesion and avoiding the coffee-ring effects.²⁹”

Comment-2: Between Lines 148-157, the manuscript would benefit from a more detailed description of the EHD printing parameters, including nozzle size, printing speed, the presence of any back pressure during printing, and other relevant operational conditions.

RESPONSE: We Thank the reviewer for valuable comments. For potential readers of the manuscript, we added more detailed description of the EHD printing parameters information in the "Results and Discussions" section of the manuscript and modified it accordingly.

The modified content can be found in **page 7, line 168-170.**

The AD-MXene ink was meticulously printed onto a SiO₂/Si wafer, employing optimized conditions to ensure consistent and stable jetting in the cone-jet mode. These precise conditions encompassed a nozzle size of 185 μm, a printing speed of 10 mm s⁻¹, a controlled flow rate of 1.3 μL min⁻¹, an applied voltage of 1.3 kV, all sustained at a working distance of 450 μm.

Comment-3: While the technical aspects of the fabrication process are well-discussed, the manuscript lacks in-depth scientific analysis. Please provide data on the conductivity of the printed MXene for different layer thicknesses, as well as surface roughness measurements. Furthermore, a detailed explanation regarding why MXene electrodes exhibit higher mobility compared to Al and Au electrodes would strengthen the scientific foundation of the paper.

RESPONSE: Thanks for the reviewer's valuable comment. As suggested by the reviewer, the electrical conductivity and surface roughness measurements for AD-MXene electrodes prepared with different number of printing cycles were performed, and the corresponding results were incorporated into the manuscript as Supplementary Figs.4d and 5.

The ADOPA ligands, which are facilitating effective charge migration between neighboring MXene nanosheets, might also favoring the effective charge carriers flow at the channel-electrode interface. Hence, AD-MXene electrodes yielded much less contact resistance as compared to conventional metal electrodes fabricated by vacuum-deposition method, which is often prone to generate defects at the channel-electrode interface. Contact resistances of TFTs with different electrodes extracted by transfer length method (Supplementary Fig. 6) confirmed low-contact resistance for AD-MXene electrodes as compared to Au and Al electrodes.

For potential readers of the manuscript, the conductivity vs number of printing cycles graph has been added as Supplementary Fig. 4d and the data corresponding to surface roughness measurements were added as Supplementary Fig. 5. In addition, the following sentences have also been added in the results and discussion and characterization sections.

The modified content can be found in page 7, line 173-176

Further, atomic force microscope (AFM) analysis revealed the gradual increase of average surface roughness (R_a) of AD-MXene lines with an increase in the number of printing cycles, and the R_a was observed to be 7.8 nm for lines fabricated with 10 printing cycles (Supplementary Fig. 5).

The added text can be found in Page 23, line 524-526

The surface morphology and roughness of AD-MXene lines were measured by AFM (XE-100, Park systems) in non-contact mode.

Revised Supplementary Fig. 4 (panel 'f' is newly added)

Supplementary Fig. 4. a-e, Cross-sectional FE-SEM images of AD-MXene lines fabricated with different number of printing cycles. f, Electrical conductivities of AD-MXene lines fabricated with 1-10 printing cycles.

Newly inserted Figure as Supplementary Fig. 5

Supplementary Fig. 5. AFM images and surface roughness of AD-MXene lines fabricated with different number of printing cycles.

Comment-4: In Lines 214-225, the manuscript reports MXene TFTs are with high mobility as well as the higher off current, but does not clarify whether this characteristic is beneficial or detrimental. Please provide a more detailed discussion on that.

RESPONSE: When fabricating a transistor with an oxide semiconductor, there are several possible reasons why the off-state current level might be high. Multiple factors can interact to result in an elevated off-state current level:

1. *Gate Leakage:* When the voltage between the gate and the source/drain decreases, gate leakage may occur. In this study, we used polymer insulating layer. While our TFT devices with the polymer insulating layer exhibited a relatively low level of gate leakage compared to other reported values from TFTs with polymer insulating layer, the off-current values were still higher. This leakage represents a situation where some current flows through the gate even when the transistor is in the off state, thereby increasing the overall off-state current level.

2. *Subthreshold Conduction:* In the case of oxide semiconductor transistors, subthreshold conduction may occur even in the off state. This phenomenon involves a small amount of current leakage, preventing complete cutoff of the current.

3. *Trap States (defects or impurities):* Oxide semiconductors may exhibit trap states (defects or impurities), where charges are captured or released. These trap states can contribute to current leakage in the off state. In our study, zinc tin oxide was used as the oxide semiconductor, where tin was doped into zinc oxide. Doping tin into zinc oxide increases the concentration of carriers within the semiconductor. Due to the increased carriers, there is a possibility of an elevation in the off-current level.

In our work, the combination of these factors seems to have increased the off current in the MXene TFTs. In fact, these complex causes can be attributed to the design of the transistor (semiconductor and dielectric patterns), the structure of the device, and its scale, rather than to the properties of the individual materials that make up the device. Minimizing these issues

in the design and manufacturing processes is crucial for reducing off-state current in the transistor.

For potential readers of the manuscript, we added more detailed description of AD-MXene based TFT operation in the "**Results and Discussions**" section of the manuscript and modified it accordingly.

The modified content can be found on page 10, line 243-247.

The off-current value of the AD-MXene based TFT was somewhat higher than that of the Au device. This was probably attributed to a combination of factors such as gate leakage, subthreshold conduction, and trap states rather than the properties of the electrode material, all of which can be sufficiently optimized by device structure design and scale control of the printed electrode.

Comment-5: There appears to be a font inconsistency at line 214. Please ensure that the font type and size are consistent throughout the manuscript for a professional and polished presentation.

RESPONSE: We thank the reviewer for valuable comment. As suggested by the reviewer, the inconsistencies in font type and size were corrected in the revised manuscript.

Comment-6: Please provide a justification for selecting PVDF-HP and FPVDF-HP as the dielectric layer. Discuss its advantages or properties that make it suitable for this application compared to other materials.

RESPONSE: We thank the reviewer for valuable comment. A similar comment was raised by Reviewer-1 as comment-1. Hence, both these comments were combinedly addressed and the modifications incorporated into the manuscript were highlighted, at our response to comment-1 of Reviewer-1. For the sake of brevity, we are not repeating the same response here.

Comment-7: In the Performance of AD-MXene as electrodes of TFTs, please include the channel length and width to aid in understanding the device configuration.

RESPONSE: Thank you to the reviewers for their valuable comments. For potential readers of the manuscript, we added more detailed description of the AD-MXene performance as electrodes in TFTs. channel length and width to aid in the "Experimental Section" section of the manuscript and modified it accordingly.

The modified content can be found on page 22, line 512-513.

Channel length and width of the ZTO layer were 236 μm and 731 μm , respectively.

Comment-8: It could be better to add the optical microscopy image of AD-MXene as electrodes of TFTs in Fig. 3.

RESPONSE: We thank the reviewer for valuable comments. As suggested by the reviewer, the optical microscopy image of AD-MXene as electrodes of TFTs in Fig. 3a is added in the revised manuscript. For your reference, the updated Figure 3 is displayed below.

Comment-9: In Fig. 4, label each component in the microscopy images to aid readers in interpreting the results.

RESPONSE: Thanks for the reviewer’s valuable comment. As suggested by the reviewer, each component in Figure 4 is labeled in the revised manuscript. The revised Figure 4 is displayed below for the reviewer reference.

Revised Figure 4:

Comment-10: The manuscript demonstrates the stability of the TFTs from FPVDF-HP but does not show how MXene contributes to this stability. Please provide evidence or a discussion that supports the role of MXene in the stability of the TFTs which makes it align with the title.

RESPONSE: Thanks for the reviewer’s valuable comment. **The AD-MXene used in this study has already been proven for its stability in our previous research. Figure R3 (*ACS nano* 2023, 17, 1112-1119) demonstrates the oxidation stability of AD-MXene.**

Figure R3a to R3d compare the oxidation stability of pristine MXene and AD-MXene in a

solution state. Over time, pristine MXene undergoes chemical oxidation facilitated by water molecules, leading to the collapse of the peak shape at around 760 nm wavelength. However, for AD-MXene dispersed in organic solvents such as EtOH and IPA, it can be confirmed that chemical oxidation does not occur even after 50 days. As illustrated in **Figure R3d**, this demonstrates that AD-MXene has excellent oxidation stability even when stored in solution for a long period of time.

Furthermore, **Figure R3e** displays the results of an 85/85 experiment (Exposure to 85% RH, 85°C) to verify the oxidation stability of AD-MXene in its film state. The difference in electrical conductivity between pristine MXene and AD-MXene films, each stored under 85/85 conditions for 20 days and subsequently dried at 150 °C to remove moisture, is clear, highlighting the excellent oxidation stability of AD-MXene.

Figure R3. Oxidation Stability Test of AD- $Ti_3C_2T_x$. UV-visible absorbance spectra of **a.** pristine $Ti_3C_2T_x$ in water, **b.** AD- $Ti_3C_2T_x$ in EtOH, and **c.** AD1- $Ti_3C_2T_x$ in IPA. **d.** The normalized intensity of UV-visible absorbance peak at ~760 nm for pristine $Ti_3C_2T_x$ in water, AD1- $Ti_3C_2T_x$ in EtOH, and AD1- $Ti_3C_2T_x$ in IPA as a function of storage time. **e.** The change of electrical conductivity of pristine $Ti_3C_2T_x$ film and AD1- $Ti_3C_2T_x$ film at 85 °C and 85% RH test condition for 20 days. After 85/85 testing, the electrical conductivity was recovered by drying at 150°C for one day. For the surface functionalization of MXene using ADOPA derivatives, ADOPA1 (3,3,4,4,5,5,6,6-

Nonafluorohexyl 2-amino-3-(3,4-dihydroxyphenyl)propanoate) and ADOPA4 (Decyl 2-amino-3-(3,4-dihydroxyphenyl)propanoate) were used.

Furthermore, we have additionally demonstrate in a recent study that when utilizing AD-MXene as a coating on practical electronic devices for conducting chemical sensor tests, the oxidation stability of AD-MXene can be substantiated (*Adv. Funct. Mater.* **2023**, 2310641). This substantiates its stability even when subjected to exposure in the surrounding ambient environment, making it suitable for use in research applications. For potential readers of the manuscript, we have addressed this information by mentioning it in the "**Results and Discussion**" section of the manuscript, made corresponding modifications, and included additional references to enhance the content.

The modified content can be found on pages 5-6, line 146-151.

Due to the enhanced hydrophobicity after ADOPA functionalization, our previous study has shown that AD-MXene is environmentally stable for extended period of time, compared to pristine waterborne MXene that is otherwise susceptible to chemical oxidation.⁴⁰ This has also been proven to be true in electronic devices with AD-MXene exposed to the surface where a recent study shows that chemical sensors based on AD-MXene retained their performance for over 6 weeks of exposure in ambient environments.⁵⁰

Comment-11: A scale bar should be inserted in Fig. 6, Fig. 7 and Fig. 9 to provide a reference for size and dimension.

RESPONSE: We thank the reviewer for valuable comment. As suggested by the reviewer, scale bars were inserted in Figure 6, Figure 7, and Figure 9. The revised figures are displayed below for your reference.

Revised Figure 6:

Revised Figure 7:

Revised Figure 9:

REVIEWERS' COMMENTS

Reviewer #1 (Remarks to the Author):

I acknowledge that authors provided appropriate clarifications to my comments on dielectric materials utilized in 1T1M device, AD-MXene concentration in ethanol, and the demonstration of ink-jet printing of AD-MXene. Further, the modifications incorporated into the revised manuscript "Functionalized MXene Ink Enables Environmentally Stable Printed Electronics" are satisfactory. Therefore, I would suggest the acceptance of this manuscript in the Nature Communications Journal without any further revisions.

Reviewer #2 (Remarks to the Author):

After the revision, my concerns have been adequately addressed. The response letter is thorough and comprehensive. I believe the current version of the manuscript is now suitable for acceptance.